# A termination-insensitive and robust electron gas at the heterointerface of two complex oxides

Meng Zhang [1,6], Kai Du[2,6], Tianshuang Ren[1], He Tian [2], Ze Zhang[2], Harold Y. Hwang[3,4] & Yanwu Xie [1,5]

The single-crystal $SrTiO_3$ (001) has two different surface terminations, $TiO_2$ and SrO. One most remarkable observation in previous studies is that only the heterointerfaces with $TiO_2$-terminated $SrTiO_3$, which usually combines with polar oxides such as $LaAlO_3$, host an electron gas. Here we show that a robust electron gas can be generated between a non-polar oxide, $CaHfO_3$, and $SrTiO_3$ (001) with either termination. Unlike the well-known electron gas of $LaAlO_3/SrTiO_3$, the present one of $CaHfO_3/SrTiO_3$ essentially has no critical thickness of $CaHfO_3$, can survive a long-time oxygen annealing at high temperature, and its transport properties are stable under exposure to water and other polar solvents. By electrostatic gating through $CaHfO_3$, field-effect devices are demonstrated using $CaHfO_3/SrTiO_3$ heterointerfaces with both terminations. These results show that the electron gas reported in the present study is unique and promising for applications in oxide electronics.

[1] Interdisciplinary Center of Quantum Information, Zhejiang Province Key Laboratory of Quantum Technology and Device, and Department of Physics, Zhejiang University, 310027 Hangzhou, China. [2] Center of Electron Microscopy, State Key Laboratory of Silicon Materials, and School of Materials Science and Engineering, Zhejiang University, 310027 Hangzhou, China. [3] Geballe Laboratory for Advanced Materials, Department of Applied Physics, Stanford University, Stanford, CA 94305, USA. [4] Stanford Institute for Materials and Energy Sciences, SLAC National Accelerator Laboratory, Menlo Park, CA 94025, USA. [5] Collaborative Innovation Center of Advanced Microstructures, Nanjing University, 210093 Nanjing, China. [6] These authors contributed equally: Meng Zhang, Kai Du. Correspondence and requests for materials should be addressed to H.T. (email: hetian@zju.edu.cn) or to Y.X. (email: ywxie@zju.edu.cn)

SrTiO$_3$ (STO) is the workhorse oxide semiconductors. Since the discovery of the high-mobility electron gas (EG) at the LaAlO$_3$/STO heterointerface[1], much effort has been undertaken to explore and understand the emergent phenomena at this heterointerface[2]. Many intriguing properties that are not present in conventional semiconductor heterostructures, such as interface magnetism[3,4], interface superconductivity[5,6], and even the coexistence of magnetism and superconductivity at the interface[7,8], have been observed. Applications in field-effect transistors (FETs)[9–11] and integrated circuit devices[12] have also been demonstrated. One most remarkable result in previous studies is that the EG can only be formed on TiO$_2$-terminated STO (T-STO), and the heterointerface with SrO-terminated STO (S-STO) is highly insulating[1,13]. This extreme asymmetry originates from the polar arrangement of atomic layers in LaAlO$_3$, and can be understood in an elegant polar discontinuity and electronic reconstruction picture[13]. Inspired by this EG, more EGs have been formed between STO and a few different polar oxides[14–19]. Following a similar idea, very recently a hole gas was achieved in a sandwiched STO/LaAlO$_3$/STO heterostructure[20,21]. In fact, nowadays interface engineering with polar oxides has become a very popular designing concept in controlling strongly correlated electrons in transition metal oxides.

However, the properties of EGs involving polar oxides can be dramatically altered by polar adsorbates[19,22], and thus suffer from ambient water[19]. Alternatively, a few studies suggested that the EGs can be generated without polar oxides (still on T-STO, when STO(001) was used)[23,24]. Particularly, they can be generated by simply capping an amorphous oxide layer on STO[24,25]. Unfortunately, EGs formed by this way are extremely fragile to oxidation environment[19]. In fact, as far as robustness is concerned, it is rare that these heterointerfaces, no matter formed with polar, non-polar, or amorphous oxides, can be comparable with LaAlO$_3$/STO because their EGs quickly vanish after annealing in oxygen at moderate temperatures, i.e. 300 °C or lower[19]. Until now, it is still a challenge to fabricate an EG that can be robust against environments including ambient water and annealing in oxygen.

In this study, we report on an EG formed at the heterointerface between epitaxial CaHfO$_3$ (CHO) and STO. Distinctly different from all the previous ones, we found that this EG can be formed on S-STO, surprisingly, as well as T-STO, and is extremely robust against environments. CHO is a non-polar wide bandgap (~6.2 eV) insulator[26]. From the ionic point of view, the cation valences of CHO is the same as that of STO since in the periodic table of the elements, Ca and Hf, as well as Sr and Ti, are respectively in the same columns. Early studies had shown that the interface between amorphous CHO and STO is metallic[27]. Here we show that the interface between epitaxial CHO and STO can be metallic as well.

## Results

### Growth and structural characterizations.
Single-crystal films of CHO were deposited on STO substrates using pulsed laser deposition. Details are described in Methods. Figure 1 shows the growth and the structural characterizations of CHO/STO of both terminations. The growth was accurately controlled by in situ monitoring reflection high-energy electron diffraction (RHEED) intensity oscillations (Fig. 1a, b). The S-STO was obtained by inserting a single unit cell (uc) of SrO on T-STO (Fig. 1b). The relatively fast decay of the RHEED intensity oscillations for depositing CHO on S-STO (Fig. 1b) indicates that the growth deviated slightly from a perfect layer-by-layer mode. However, this imperfection does not affect the quality of CHO significantly. The atomic force microscopy (AFM) images of the surface of CHO/

STO of both terminations (Fig. 1c, d) show the steps and terraces of the original substrate surface, indicating high-quality growth. The high quality of the CHO films can also be appreciated from the X-ray diffraction (XRD) finite-thickness oscillations (Fig. 1e, f). The XRD data (Fig. 1e, f, and Supplementary Fig. 1) reveal epitaxial growth and very high crystallinity. The high-angle annular dark-field scanning transmission electron microscopy (HAADF-STEM) images reveal that the CHO films are coherent with the STO substrates for both terminations (Fig. 1g, h), and the two types of heterointerfaces can be distinguished by the atomic-scale energy-dispersive X-ray spectroscopy (EDS) elemental mapping (Fig. 1i, j).

### Electrical transport properties.
Figure 2a, b shows the temperature dependence of the sheet resistance $R_{sheet}$ for the as-grown CHO/STO heterointerfaces with the thickness of CHO varying from 1 to 30 uc. One important observation is that, unlike all the known heterointerfaces such as LaAlO$_3$/STO[1,13], in which only those prepared on the T-STO show interfacial conductivity, the CHO/STO heterointerfaces of both terminations show very high conductivity. Our control experiments demonstrated that the heterointerface of LaAlO$_3$/S-STO prepared under the same growth condition as CHO/S-STO is highly insulating (see Supplementary Fig. 2), and the CHO/STO heterointerface grown on unterminated STO substrate is still conducting (see Supplementary Fig. 3). These observations confirm that the observed conducting interface of CHO/S-STO is not caused by any incompleteness of the SrO coverage. We also confirm that the STO substrates underwent the same growth condition, but without film growth and post annealing are highly insulating (beyond our measurement limit). Hall effect measurement reveals that for CHO/STO of both terminations, the mobile carriers are electrons (see Supplementary Fig. 4). With increasing the thickness of CHO, $R_{sheet}$ shows a general, although not monotonic, trend to decrease (see Fig. 2a, b and Supplementary Fig. 5a, b). Similar observation was reported on CaZrO$_3$/STO previously[16]. We found that this trend becomes less obvious for the annealed samples (see Supplementary Fig. 5c, d), suggesting that it is related to the CHO-thickness-dependent amount of oxygen vacancies in STO.

Another interesting observation is that, for CHO/STO with both terminations, although their conductivities change with the thickness of CHO, there is still measurable conductivity at room temperature even when the thickness of CHO is as thin as 1 uc (see Fig. 2a, b and Supplementary Figs. 6a, b and 7). We found that the room-temperature conductivity of 1 uc CHO/STO is remarkably stable and showed no obvious aging after 1-year storage. By contrast, when the thickness of LaAlO$_3$ is 3 uc or below, the LaAlO$_3$/STO heterointerface is highly insulating (see Supplementary Fig. 2)[28]. Therefore, there is no strict critical thickness in CHO/STO. These two observations indicate that the EG observed at the CHO/STO heterointerfaces is different from all the previous ones[1,15–19,23].

The EG of CHO/STO is robust against annealing in oxygen (Fig. 2c–f, Supplementary Figs. 6 and 7). Previous studies[19,29] had shown that the electrical conductivity and behavior of all the oxide EGs can be strongly suppressed by annealing in oxygen, because annealing will remove oxygen vacancies in both the STO substrates and the films. Most previously reported EGs cannot survive an annealing in oxygen at 300 °C or even much lower[19]. In contrast, Fig. 2c, d shows that the CHO/STO can remain metallic over the whole temperature range (or above 100 K) after an in situ annealing at a temperature of 500 °C (or 600 °C), under partial pressure $P(O_2) = 200$ mbar. This robustness is close to that of LaAlO$_3$/T-STO. Furthermore, we found that even after a 24-h

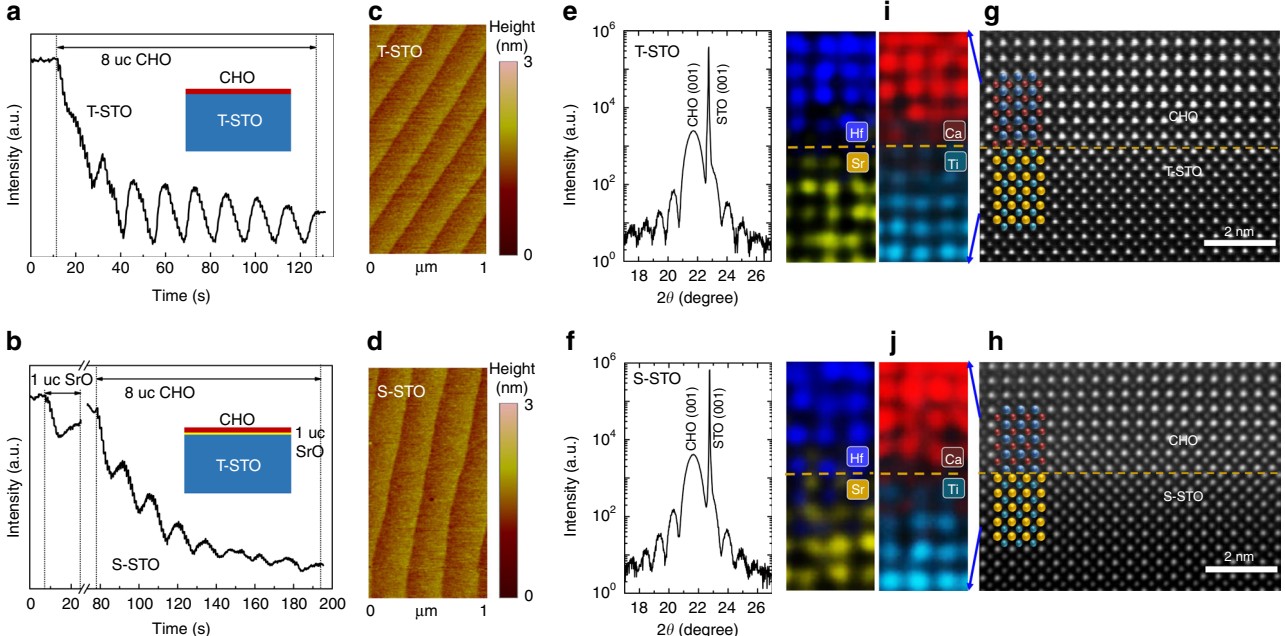

**Fig. 1** Growth and structural characterizations of two types of interfaces between CaHfO₃ (CHO) and SrTiO₃ (STO). **a**, **b** Reflection high-energy electron diffraction (RHEED) intensity oscillations of the specular reflected beam for the growth of CHO directly on TiO₂-terminated STO (T-STO), and after a monolayer of SrO was deposited on the T-STO, namely SrO-terminated STO (S-STO), respectively. Insets show the schematics of the two structures. **c**, **d** Atomic force microscopy (AFM) surface morphologies of 8 uc CHO grown on T-STO and S-STO, respectively. The size of both images is 1 μm × 2 μm. **e**, **f** X-ray diffraction (XRD) finite-thickness oscillations in the vicinity of the (001) reflection for 20 uc CHO grown on T-STO and S-STO, respectively. **g**, **h** Cross-sectional scanning transmission electron microscopy (STEM) images of 8 uc CHO grown on T-STO and S-STO, respectively. **i**, **j** The energy-dispersive X-ray spectroscopy (EDS) elemental mapping of the same samples from **g**, **h**, respectively

annealing at 1000 °C, under $P(O_2) = 1$ bar, the CHO/STO can still remain metallic down to 150 K (Fig. 2e, f). Note that our control experiments showed that even LaAlO₃/T-STO became highly insulating after the same extreme annealing (see Supplementary Fig. 9).

Besides its robustness to the annealing in oxygen, the presently observed EG of CHO/STO shows a few other interesting features. First, the CHO/S-STO generally has even better electrical transport properties than the CHO/T-STO (Fig. 2). Second, the electron mobility of CHO/STO is high. As shown in Fig. 2 and Supplementary Figs. 4, 6, and 7, for the metallic CHO/STO, the residual resistivity ratio ($R_{sheet}$(300 K)/$R_{sheet}$(2 K)) can be more than three orders of magnitude, indicating a very high mobility. Hall effect measurement reveals that the low-temperature mobility can reach 10,000 cm²/V/s (Supplementary Fig. 4c), comparable with that of the best prepared LaAlO₃/T-STO samples[30]. Third, unlike LaAlO₃/T-STO[19,22], the EG of CHO/STO is extremely stable against surface exposure of polar solvents. We treated the CHO/STO samples, prepared under various conditions, with water and other polar solvents as we did previously for EGs at other heterointerfaces[19,22], and found negligible influence on their conductivity (see Supplementary Fig. 10 and Supplementary Table 1). This point is important for device applications because it assures that the performance of devices can be stable under different environments.

**Possible origin of the EG of CHO/STO**. We now consider the possible origin of the EG of CHO/STO. Since CHO, unlike LaAlO₃, is non-polar, one can exclude the classic polar discontinuity and electronic reconstruction scenario[13]. An alternative plausible scenario is the strain-induced polarization and electronic reconstruction, as proposed by a theoretical study[31] and experimentally observed at the heterointerface of CaZrO₃/STO[16]. However, the B-site atom displacement vector maps of

CHO/STO (see Supplementary Fig. 11) show that the atom displacements are tiny and random, and the resulting net out-of-plane polarization is within the detection error of our STEM measurement. Furthermore, the absence of a critical thickness and the inertness to polar solvents are unexpected from a strain-induced polarization mechanism. Therefore, we conclude that the EG of CHO/STO is unlikely to originate from the strain-induced polarization in CHO films.

One other possible cause is the oxygen non-stoichiometry. Experimental data show that the overall conductivity decreases with increasing partial oxygen pressure during growth (see Supplementary Fig. 8), and the overall conductivity increases with increasing thickness of CHO (Fig. 2a, b and Supplementary Fig. 5). These findings suggest that oxygen non-stoichiometry in the CHO films (rather than the STO substrate) plays an important role in the formation of EG. The EG may be formed due to the oxygen non-stoichiometry in the CHO films, by transferring electrons to the interfacial STO[19,32,33] or by generating oxygen vacancies in the interfacial STO via oxygen migration[34,35]. A remaining puzzle is why the EG of CHO/STO can survive such a severe annealing in oxygen. One interesting possibility is that the existence of some oxygen vacancies in CHO or the interface of CHO/STO is thermodynamically stable, similar to the cases of the surface of LaAlO₃/T-STO[32], the interface of LaAlO₃/S-STO[13], or the interface of γ-Al₂O₃/T-STO[23].

**Device application**. CHO has an orthorhombic structure with a pseudocubic lattice constant of 3.990 Å; STO has a cubic lattice constant of 3.905 Å. The good matching of their lattices makes CHO a natural gating and insulating material for STO-based oxide devices[36]. Early studies from Lippmaa group[37,38] had shown that both the crystalline and the amorphous CHO films grown on STO exhibit excellent dielectric properties with a relative dielectric constant around 17, comparable with LaAlO₃ or

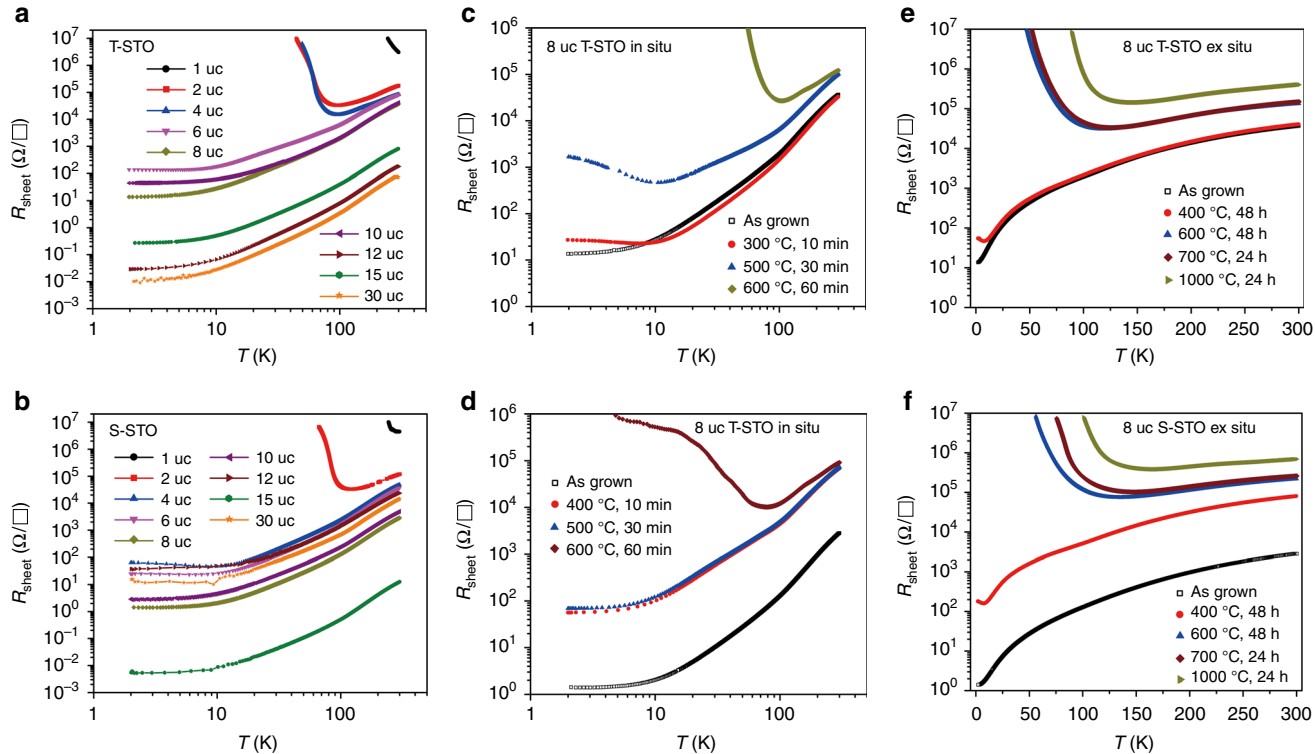

**Fig. 2** Electrical transport properties of CaHfO₃/SrTiO₃ (CHO/STO) heterointerfaces. **a, b** Temperature dependence of $R_{sheet}$ for the as-grown CHO of various thicknesses grown on TiO₂-terminated STO (T-STO) and SrO-terminated STO (S-STO), respectively. The same samples were post-annealed ex situ in O₂, and the corresponding results are shown in Supplementary Fig. 6. **c, d** Temperature dependence of $R_{sheet}$ for the in situ annealed CHO (8 uc) grown on T-STO and S-STO, respectively. Each curve represents a sample that has been annealed in situ under $P(O_2) = 200$ mbar, at the conditions as labeled. More in situ annealing results on samples of different thicknesses are presented in Supplementary Fig. 7. **e, f** Temperature dependence of $R_{sheet}$ for the ex situ annealed CHO (8 uc) grown on T-STO and S-STO, respectively. In each case, one as-grown sample was annealed ex situ subsequently from low to high annealing temperature, under $P(O_2) = 1$ bar, in a tube furnace. For clarity, double-logarithmic coordinates were used in **a–d**, and semi-logarithmic coordinates were used in **e, f**

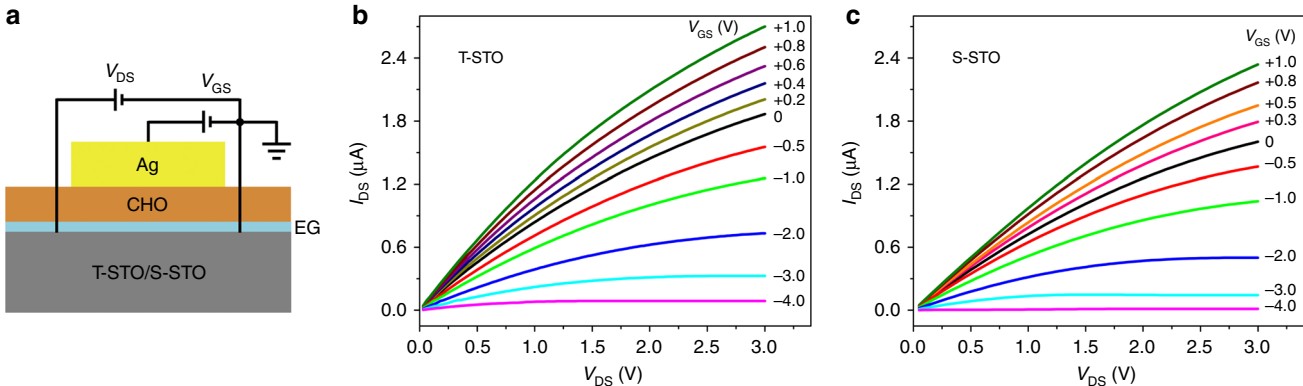

**Fig. 3** Field-effect transistors with in situ annealed 8-uc CaHfO₃/SrTiO₃ (CHO/STO) heterointerfaces. **a** Sketch of a device. The channel is 10 μm in width and 100 μm in length. The top gate electrode is a 100-nm-thick silver film. Drain-source current ($I_{DS}$) versus drain-source voltage ($V_{DS}$) at different gate-source voltages ($V_{GS}$) for **b** CHO grown on TiO₂-terminated STO (T-STO) and **c** SrO-terminated STO (S-STO). All data were measured at room temperature

HfO₂ (note that HfO₂ is replacing SiO₂ as the gating and insulating material in commercial integrated circuits). Here, we demonstrate that the CHO/STO of both terminations can be used to fabricate field-effect devices.

As schematically shown in Fig. 3a, the EG at the CHO/STO heterointerface is used as the drain (D)-source (S) channel, and the epitaxial CHO film as gate dielectrics, for FET operations. The output characteristics, drain-source current ($I_{DS}$) versus drain-source voltage ($V_{DS}$) at different gate voltages ($V_{GS}$), are shown in

Fig. 3b, c. Note that although the thickness of CHO is only 8 uc (~3.2 nm) (similar FETs with 15 uc CHO are shown in Supplementary Fig. 13), in the $-4 \text{ V} < V_{GS} < 1 \text{ V}$ range, the leakage current of the devices is very small (see Supplementary Fig. 12). In addition, we found that even increasing the size of the contact between the gate electrode and the CHO film up to several millimeter square, in the same $V_{GS}$ range the leakage current remains small (not shown), confirming that the CHO film of CHO/STO is very homogeneous.

## Discussion

The present study demonstrates that a robust EG can be generated at the heterointerface of an epitaxially grown CHO film and a STO substrate with either termination. This EG is distinctly different from that of LaAlO$_3$/T-STO because it is insensitive to the termination of STO, has no strict critical thickness of CHO, and is stable against polar adsorbates. This EG is also remarkably stable against annealing in oxygen, comparable with that of LaAlO$_3$/T-STO. Using this EG as the conducting channel, and CHO as gating dielectrics, FETs are constructed. The high-performance gating ability, integrated with the robustness to environments (temperature, oxygen, humidity, and solvents), makes the EG at the interfaces of CHO/STO very attractive for applications in oxide electronics.

## Methods

**Material**. CHO has an orthorhombic structure with $a = 5.719$, $b = 7.982$, and $c = 5.5578$ Å[39]. The pseudocubic lattice constant is 3.990 Å. STO has a cubic lattice constant of 3.905 Å. The average lattice mismatch between them is $+2.3\%$[39]. The XRD reciprocal space maps (Supplementary Fig. 1c, d) indicate that for both terminations of STO, the CHO films are constrained on the STO substrates, with an in-plane lattice constant of 3.905 Å and out-of-plane lattice constant of 4.075 Å for CHO/T-STO, and 4.102 Å for CHO/S-STO.

**Pulsed laser deposition synthesis**. T-STO substrates were pre-annealed in situ at an oxygen partial pressure $P(O_2)$ of $1 \times 10^{-4}$ mbar for 20 min at a temperature of 975 °C to achieve sharp step-and-terrace surfaces. S-STO substrates were prepared by depositing 1 uc SrO layer on the pre-annealed T-STO substrates in situ at a temperature of 800 °C, and $P(O_2) = 1 \times 10^{-4}$ mbar, using 1.5 J/cm$^2$ laser fluence. A 248-nm KrF excimer laser was used for the deposition. All the CHO films presented in this work were grown at a substrate temperature $T_g$ of 800 °C, and $P(O_2) = 1 \times 10^{-4}$ mbar, using 3 J/cm$^2$ laser fluence. After growth, the samples without in situ post-annealing were cooled down at a rate of 100 °C/min under the growth oxygen pressure; the samples with in situ post-annealing were cooled down at a rate of 100 °C/min to 500 °C, stayed at 500 °C for 30 min under $P(O_2) = 200$ mbar, and then cooled down to room temperature under the same oxygen pressure. The CHO polycrystalline target was prepared by sintering a mixture of stoichiometric amounts of CaCO$_3$ and HfO$_2$ at 1300 °C for 12 h.

**Structural characterizations**. The AFM images were acquired in tapping mode with Veeco DI3100. The XRD data were taken using a monochromated Cu-K$_\alpha$ source on a Bruker AXS D8-Discover. Cross-sectional specimens for STEM investigations were prepared by a FEI Quanta 3D FEG Focused Ion Beam. STEM images were acquired using a spherical aberration-corrected microscope equipped with four Super-X EDS detectors (FEI Titan G2 80-200 Chemi STEM, 30 mrad convergence angle).

**Electrical transport measurement**. The conducting CHO/STO interfaces were contacted by ultrasonic bonding with Al wires. The conductivity measurements were performed using a standard four-probe method. The Hall effect measurements were performed using a Hall-bar sample made with Al$_2$O$_x$ hard mask[30].

## Data availability

The data that support the findings of this study are available from the corresponding authors upon reasonable request.

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

## Acknowledgements

We greatly thank Dr. Hongrui Zhang, Prof. Jirong Sun, and Prof. Baogen Shen for the help in XRD measurements. This work was supported by the National Key R&D Program of the MOST of China (Grants No. 2016YFA0300204, No. 2017YFA0303002), and the Fundamental Research Funds for the Central Universities.

## Author contributions

M.Z. and T.R. fabricated and characterized the heterointerfaces. K.D., H.T. and Z.Z. measured and analyzed the STEM data. H.Y.H. guided the work. Y.X. designed the experiment and wrote the manuscript, with input from all authors.

## Additional information

**Competing interests:** The authors declare no competing interests.

