## [Peer Review File · Nature Communications]

Reviewers' comments:

Reviewer #1 (Remarks to the Author):

This manuscript reports on the creation of an electron gas at the CaHfO₃/SrTiO₃ interface that is independent of the termination of the SrTiO₃ termination. This electron system is also remarkably robust against oxidation and chemical treatment. This is important news to the field, and of possible relevance for applications. As the paper is also very well written, I recommend it for publication.

In addition, I have the following suggestions to the authors:

- 1) the paper would benefit if the authors presented an analysis of the completeness of the SrO coverage and also informed on whether conducting interfacial layers could be grown even on unterminated substrates.
- 2) in the methods section: please also inform about the cooldown procedure employed after the CaHfO₃ growth. Also, please present the preparation of the SrO-terminated substrates in the methods section.
- 3) If appropriate, add "at room temperature" (or so) to the sentence "... there is still measurable conductivity even when the thickness of CHO is as thin as 1 uc."

Reviewer #2 (Remarks to the Author):

Zhang et al present evidence that the heterointerface of CaHfO₃/SrTiO₃ supports a conducting electron gas, continuing a trend in recent years of investigating novel oxide heterostructures. What makes CHO/STO potentially interesting are its insensitivity to STO surface termination, no critical thickness of CHO, and insensitivity to chemicals on the surface. These results are potentially very interesting and intriguing; however, I think the results as presented are not yet suitable for publication in Nature Communications.

1. Figure 1. Why is the RHEED oscillations not as strong for SrO-terminated STO? The STEM indicates that the epitaxy is just as good as TiO₂-terminated STO. No discussion is given.
2. Figure 2. The thickness dependence is difficult to understand without a graph of R_s vs thickness. For example, is there any monotonic dependence of R_s on thickness? Or mobility, etc? It appears that R_s continues to decrease when the thickness of CHO is increased, which seems to be a peculiar result. More investigation or discussion is needed.
3. Page 6. Discussion of annealing alludes to control experiments with LAO/STO, but the results are not shown. I think it would be helpful for the control experiments to be shown.
4. Page 7. The authors discuss adding water and polar solvents, but little details or data are shown. For example, for what thicknesses of CHO were tried?
5. Figure 3. Why were FETs fabricated only on 8uc samples? 15 uc has much lower R_s and therefore higher mobility and therefore would seem to have better device performance.
6. Figure S2. Unusual minimum in R_s at 20K is not discussed.

Reviewer #3 (Remarks to the Author):

Dear Editor,

The manuscript of Zhang et al. reports a new two-dimensional electron gas (2DEGs) at the interface between two oxide insulators of CaHfO₃ and SrTiO₃, the latter is supposed to be the foundation of oxide electronics. The new 2DEG system exhibit two remarkable properties: a relatively high mobility and an extreme stability upon oxygen annealing. The mechanism underlying the interface conduction which points to the interface stabilized oxygen vacancies is

also reasonable. The new system also shows promising device performance as field effect transistors. I view this manuscript is timing and very interesting. This is because most of the research on conducting oxide interfaces has so far limited to the polar/nonpolar system of LaAlO₃/SrTiO₃, other systems such as the nonpolar/nonpolar systems as investigated here have rarely been explored. More importantly, identifying a system which is robust to oxygen annealing and surface dopants has significant importance for practical application. Therefore, I recommend the publication of this manuscript for Nature communications.

There are some minor issues which may further improve the manuscript:

1. As shown in Fig.2, the carrier density of the CaHfO₃/SrTiO₃ system depends strongly on the film thickness. Is the relationship between the electron mobility and the carrier density fulfil the universality of electron mobility found in LaAlO₃/SrTiO₃ system, see for example Trier et al. Appl. Phys. Lett. 111, 092106 (2017)?
2. What is the lattice mismatch between the CaHfO₃ and SrTiO₃, and what is the lattice parameter of the CaHfO₃ films?
3. For the 1 μ c sample, which is nonmetallic, is there any aging effect? In other words, is it still conductive after one week or is there any photoconductivity contribution to the conduction?
4. If the STO undergoes the growth condition (high temperature of 800 degreeC, and oxygen pressure of 1E-4 mbar), without film growth and post annealing, will it become metallic or insulating? This has to be addressed in the main text.
5. It is notably to mention that similar thermodynamically stable oxygen vacancies have also been observed in gamma-Al₂O₃/STO interface as discussed in Ref. 23.

Reviewer #4 (Remarks to the Author):

The work presented in this manuscript shows yet another aspect of the complexity of the interfaces between different metal oxide thin films and SrTiO₃ (STO) substrates of different terminations. The manuscript is well written (although definite articles are missing at some places, and minor typos exist) and is easy to follow. Materials- and applications wise, the CaHfO₃/STO interface has been studied before (with epitaxial CaHfO₃!) in other contexts (references are also quoted in the manuscript, e.g., ref 38), so the material combination is not new. The main differences to previous works of other metal oxide interfaces to STO is the observation that for CaHfO₃/STO, not only does the TiO₂ terminated STO show high conductivity and mobility, but also the SrO terminated STO (other publications have shown an insulating interface for the SrO terminated STO with LaAlO₃ overlayers) and that there is no critical thickness for the CaHfO₃ for the electron gas to be created for both the TiO₂- and SrO-terminated STO. However, in previous works the critical transition for the metal-insulator transition between 3 and 4 unit cells of overlying material were observed with in-situ post-annealing in O₂, which seems not to have been done in this work (Fig. 2a-b). Therefore, the absence of the metal-insulator transition may not be directly comparable to previous findings. The authors present some of the possible reasons for the interface conductivity and claims that the electron gas is of a new nature. However, there is no evidence for that the CaHfO₃/STO interface electron gas has different properties from previous studies, with the exception that also the SrO termination is conductive, which could be caused by oxygen vacancies in the absence of the post-annealing process. (The scenario of oxygen vacancies being responsible for the metallicity is theoretically corroborated ref. 35.) The authors further claim that "...an experimental observation of EG at the interface between epitaxial CHO film and STO has never been reported.", while in ref. 38, field induced conductivity was reported for epitaxial CaHfO₃ grown at somewhat lower temperature (700°C instead of 800°C) and higher

oxygen pressure (1mbar instead of 10^{-4} mbar), and lower laser fluency. The direct comparison would imply that the higher oxygen pressure in ref. 38 might have mitigated the oxygen vacancy problem while that interface gets conducting only by gating. In conclusion, I see the main benefits of the presented work to be an additional observation of what has been reported before, but now also with SrO-terminated STO (although for different growth conditions) and a claimed robustness of the interface. For this reason, I would recommend that the paper would find a more appropriate journal in e.g., Journal of Applied Physics or Applied Physics Letters.

Additional comments to the text:

p.2 l.1: "SrTiO₃ (STO) is the workhorse oxide semiconductor." Maybe write "SrTiO₃ (STO) is the oxide semiconductor workhorse." Or "SrTiO₃ (STO) is the workhorse of oxide semiconductors."

p.2 l.6: Maybe write "...superconductivity at the interface..."

p.2 l.19: Maybe write "that the EGs can be generated..."

p.4 Fig. 1: l.6 Maybe write "g, h, Cross-sectional STEM images of 8 uc..."

p.9 Fig. 3: Maybe write "The top gate electrode is a 100-nm-thick silver film." (Misspelled "top" and missing "a".)

p.9 l.14-15: Maybe write "...to achieve sharp step-and-terrace surfaces" or "...to achieve a sharp step-and-terrace surface"

Check also page reference of ref. 35 (Appl. Phys. Lett. 98, 133114 (2011))!

Response to Reviewer #1:

General comment: “This manuscript reports on the creation of an electron gas at the $\text{CaHfO}_3/\text{SrTiO}_3$ interface that is independent of the termination of the SrTiO_3 termination. This electron system is also remarkably robust against oxidation and chemical treatment. This is important news to the field, and of possible relevance for applications. As the paper is also very well written, I recommend it for publication.”

Response: We thank the referee for the recommendation for publication. The point-by-point response to the referee’s suggestions is in the following.

Point 1. “In addition, I have the following suggestions to the authors:

1) the paper would benefit if the authors presented an analysis of the completeness of the SrO coverage and also informed on whether conducting interfaces could be grown even on unterminated substrates.”

Response: We thank the referee for this suggestion. In the revised manuscript, we added a short paragraph (page 5) of analysis “Our control experiments demonstrated that the heterointerface of $\text{LaAlO}_3/\text{S-STO}$ prepared under the same growth condition as $\text{CHO}/\text{S-STO}$ is highly insulating (see Fig. S2), and the CHO/STO heterointerface grown on unterminated STO substrate is conducting (see Fig. S3). These observations confirm that the observed conducting interface of $\text{CHO}/\text{S-STO}$ is not caused by any incompleteness of the SrO coverage.”

Figs. S2 & S3 are presented here as Figs. R1 & R2, respectively.

Fig. R1 (see the red triangle symbols, 8 uc $\text{LaAlO}_3/\text{S-STO}$)

Fig. R2 | 8 uc CHO grown on unterminated STO .

Point 2. “2) in the methods section: please also inform about the cool down procedure employed after the CaHfO₃ growth. Also, please present the preparation of the SrO-terminated substrates in the methods section.”

Response: We thank the referee for this suggestion. In the revised manuscript, we added them in the Methods section.

Page 11: “After growth, the samples without *in-situ* post annealing were cooled down at 100 °C/min under the growth oxygen pressure; the samples with *in-situ* post annealing were cooled down at 100 °C/min to 500 °C, stayed at 500 °C for 30 minutes under P(O₂) = 200 mbar, and then cooled down to room temperature under the same oxygen pressure.”

Page 10: “S-STO substrates were prepared by depositing 1-uc SrO layer on the pre-annealed T-STO substrates *in situ* at 800 °C, and P(O₂) = 1×10⁻⁴ mbar, using 1.5 J/cm² laser fluence.”

Point 3. “3) If appropriate, add "at room temperature" (or so) to the sentence"... there is still measurable conductivity even when the thickness of CHO is as thin as 1 uc.””

Response: We thank the referee for this suggestion. In the revised manuscript (page 5), we added “at room temperature” to the sentence accordingly.

Response to Reviewer #2:

General comment. “Zhang et al present evidence that the heterointerface of CaHfO₃/SrTiO₃ supports a conducting electron gas, continuing a trend in recent years of investigating novel oxide heterostructures. What makes CHO/STO potentially interesting are its insensitivity to STO surface termination, no critical thickness of CHO, and insensitivity to chemicals on the surface. These results are potentially very interesting and intriguing; however, I think the results as presented are not yet suitable for publication in Nature Communications.”

Response: We thank the referee for the positive comments about our new findings on CHO/STO. Following the referee’s suggestions, we have carried out more experiments and analyses. In the revised manuscript, we provided new data and discussion, and made the presentation clearer. The point-by-point response to the referee’s concerns is in the following.

Point 1. “1. Figure 1. Why is the RHEED oscillations not as strong for SrO-terminated STO? The STEM indicates that the epitaxy is just as good as TiO₂-terminated STO. No discussion is given.”

Response: We thank the referee for pointing out this issue. The RHEED oscillations are very sensitive to many factors. One possibility is that the pre-deposition of SrO changed the surface diffusion ability of species from the PLD plume, which made the growth of CHO deviate slightly from the perfect layer-by-layer mode. In the revised manuscript, we added a short paragraph (page 3) to discuss this issue “The relatively fast decay of the RHEED intensity oscillations for depositing CHO on S-STO (Fig. 1b) indicates that the growth deviated slightly from a perfect layer-by-layer mode. However, this imperfection does not affect the quality of CHO significantly.”

Point 2. “2. Figure 2. The thickness dependence is difficult to understand without a graph of R_s vs thickness. For example, is there any monotonic dependence of R_s on thickness? Or mobility, etc? It appears that R_s continues to decrease when the thickness of CHO is increased, which seems to be a peculiar result. More investigation or discussion is needed.”

Response: We thank the referee for pointing out this issue. In the revised manuscript, we provided graphs of R_s vs thickness at different temperatures for both the as-grown and the annealed samples (see Figs. S5a-d). A short discussion (page 5) was given accordingly: “With increasing the thickness of CHO, R_{sheet} shows a general, although not monotonic, trend to decrease (see Figs. 2a-b & S5a-b). Similar observation was reported on $\text{CaZrO}_3/\text{STO}$ previously¹⁶. We found that this trend becomes less obvious for the annealed samples (see Fig. S5c-d), suggesting that it is related to the CHO-thickness-dependent amount of oxygen vacancies in STO.”

The dependence of the amount of oxygen vacancies in STO on the thickness of CHO might be explained as following. During growth oxygen vacancies were created in the CHO film. At the growth condition (high temperature and low oxygen pressure), these oxygen vacancies can move into STO substrates by migration. One would expect that more oxygen vacancies will move into STO for the thicker CHO because the total amount of oxygen vacancies is larger in thicker CHO.

Fig. S5 is presented here as Fig. R3.

Fig. R3 | R_s vs CHO thickness. [A] denotes the results from the annealed samples.

Point 3. “3. Page 6. Discussion of annealing alludes to control experiments with LAO/STO, but the results are not shown. I think it would be helpful for the control experiments to be shown.”

Response: We thank the referee for this suggestion. In the revised manuscript, we provided the results of the control annealing experiments with LAO/STO (see Fig. S9).

Fig. S9 is presented here as Fig. R4.

Fig. R4 | Annealing LAO/STO under extreme condition. Although LAO/STO showed an even better conductivity after annealing at 700 °C, it become fully insulating after annealing at 1000 °C.

Point 4. “4. Page 7. The authors discuss adding water and polar solvents, but little details or data are shown. For example, for what thicknesses of CHO were tried?”

Response: We thank the referee for this suggestion. In the revised manuscript, we provided more details and data about the experiments of adding water and polar solvents (see Fig. S10 & Table S1). Fig. S10 and Table S1 are presented here as Fig. R5 and Table R1, respectively.

Fig. R5 | Effect of surface treating with DI water. The thickness of LaAlO₃ and CHO is 8 uc.

samples change ratio (%) polar solvents	LaAlO ₃ /T-STO	CHO/T-STO	CHO/S-STO
DI-water	-100.5%	-5.7%	-6.7%
ethanol	-206.5%	-7.7%	-6.6%
acetone	-209.7%	-7.5%	-5.6%

Table R1 | Effect of surface treating with different polar solvents. The thickness of LaAlO₃ and CHO is 8 uc. The data were collected at room temperature. The change ratio is defined

as $[R_{\text{sheet}}(\text{polar}) - R_{\text{sheet}}(\text{initial})]/R_{\text{sheet}}(\text{polar})$.

Point 5. “5. Figure 3. Why were FETs fabricated only on 8uc samples? 15 uc has much lower R_s and therefore higher mobility and therefore would seem to have better device performance.”

Response: We thank the referee for this suggestion. Following the referee’s suggestion, in the revised manuscript, we added the results of 15 uc samples as well (see Fig. S13). These new results are comparable with those of 8 uc samples.

Fig. S13 is presented here as Fig. R6.

Fig. R6 | Field-effect transistor with CHO (15 uc)/STO heterointerfaces. a & b, for T-STO. c & d, for S-STO. The data were collected at room temperature.

Point 6. “6. Figure S2. Unusual minimum in R_s at 20K is not discussed.”

Response: We thank the referee for pointing out this concern. In the revised manuscript, we presented a short discussion in the caption of Fig. S4 (the original Figure S2) to address this issue. “The low-temperature upturn of the $R_{\text{sheet}}-T$ curve for the 8 μm CHO/T-STO might be attributed to Kondo scattering. Similar observations have been reported in $\text{LaAlO}_3/\text{STO}^{1-3}$ and electrolyte gated STO^4 . ”

Response to Reviewer #3:

General comment. “The manuscript of Zhang et al. reports a new two-dimensional electron gas (2DEGs) at the interface between two oxide insulators of CaHfO_3 and SrTiO_3 , the latter is supposed to be the foundation of oxide electronics. The new 2DEG system exhibit two remarkable properties: a relatively high mobility and an extreme stability upon oxygen annealing. The mechanism underlying the interface conduction which points to the interface stabilized oxygen vacancies is also reasonable. The new system also shows promising device performance as field effect transistors. I view this manuscript is timing and very interesting. This is because most of the research on conducting oxide interfaces has so far limited to the polar/nonpolar system of $\text{LaAlO}_3/\text{SrTiO}_3$, other systems such as the nonpolar/nonpolar systems as investigated here have rarely been explored. More importantly, identifying a system which is robust to oxygen annealing and surface dopants has significant importance for practical application. Therefore, I recommend the publication of this manuscript for Nature communications.”

Response: We thank the referee for the recommendation for publication. The point-by-point response to the referee’s concerns is in the following.

Point 1. “There are some minor issues which may further improve the manuscript:

1. As shown in Fig. 2, the carrier density of the $\text{CaHfO}_3/\text{SrTiO}_3$ system depends strongly on the film thickness. Is the relationship between the electron mobility and the carrier density fulfil the universality of electron mobility found in $\text{LaAlO}_3/\text{SrTiO}_3$ system, see for example Trier *et al.* Appl. Phys. Lett. 111, 092106 (2017)?”

Response: We thank the referee for this suggestion. We compared our data with the data shown in the reference of Trier *et al.* The result is as shown in Fig. R7.

Fig. R7. The background figure is a photo grasp of FIG. 1 in the reference of Trier *et al.* The green circles represent the data of our CHO/STO samples.

As shown in Fig. R7, in CaHfO₃/SrTiO₃ heterointerfaces the electron mobility shows a less strong dependence on the sheet carrier density. Since the detailed study of this issue is beyond the scope of the present work, we would like to leave it for future studies.

Point 2. “2. What is the lattice mismatch between the CaHfO₃ and SrTiO₃, and what is the lattice parameter of the CaHfO₃ films?”

Response: We thank the referee for pointing out this issue. In the revised manuscript, we added a paragraph in the Methods section (page 10) to state these details.

“Material. CHO has an orthorhombic structure with $a=5.719$, $b=7.982$, and $c=5.5578$ Å³⁹. The pseudocubic lattice constant is 3.990 Å. SrTiO₃ has a cubic lattice constant of 3.905 Å. The average lattice mismatch between them is +2.3%³⁹. The XRD reciprocal space maps (Figs. S1c and S1d) indicate that for both terminations of STO, the CHO films are constrained on the STO substrates, with an in-plane lattice constant of 3.905 Å and out-of-plane lattice constant of 4.075 Å for CHO/T-STO, and 4.102 Å for CHO/S-STO. ”

Point 3. “3. For the 1 uc sample, which is nonmetallic, is there any aging effect? In other words, is it still conductive after one week or is there any photoconductivity contribution to the conduction?”

Response: We thank the referee for reminding us this good point. We have compared the room-temperature conductivities of CHO (1 uc)/STO of both terminations when they were just prepared and after one-year storage. We found that the samples are still conductive after one year from their fabrications. We also measured these samples in dark and the results were nearly the same, suggesting that the contribution of photoconductivity is negligible.

In the revised manuscript, we added a short statement (page 6) to address this point “We found that the room-temperature conductivity of 1 uc CHO/STO is remarkably stable and showed no obvious aging after one-year storage.”

Point 4. “4. If the STO undergoes the growth condition (high temperature of 800 degree C, and oxygen pressure of 1E-4 mbar), without film growth and post annealing, will it become metallic or insulating? This has to be addressed in the main text.”

Response: As suggested by the referee, we carefully prepared STO that underwent the same growth condition, without film growth and post annealing. We found that they are highly insulating (beyond our measurement limit). In the revised manuscript, we addressed this point clearly (page 5) “We also confirm that the STO substrates underwent the same growth condition, but without film growth and post annealing, are highly insulating (beyond our measurement limit).”.

Point 5. “5. It is notably to mention that similar thermodynamically stable oxygen vacancies have also been observed in gamma-Al₂O₃/STO interface as discussed in Ref. 23.”

Response: We thank the referee for reminding us this point. In the revised manuscript, we added an additional citation to Ref. 23 when discussing the mechanism of the origin of the electron gas.

Response to Reviewer #4:

General comment. “The work presented in this manuscript shows yet another aspect of the complexity of the interfaces between different metal oxide thin films and SrTiO₃ (STO) substrates of different terminations. The manuscript is well written (although definite articles are missing at some places, and minor typos exist) and is easy to follow.”

Response: We thank the referee for this positive comment about our manuscript. We also thank the referee for the critical comments which we will address point-by-point below.

Point 1. “Materials- and applications wise, the CaHfO₃/STO interface has been studied before (with epitaxial CaHfO₃!) in other contexts (references are also quoted in the manuscript, e.g., ref 38), so the material combination is not new.”

Response: We agree with the referee that materials- and applications wise, the CaHfO₃/STO heterostructure has been studied before, and so the *material combination* is not new. However, we would like to stress that the motivations, experiments, results, key findings, and conclusions of the previous studies are *significantly different* from ours. In order to better address this comment of the referee, please allow us give a short summary of the previous studies first, and then figure out why our study is substantially different from the previous ones.

The experimental pioneering studies of CaHfO₃/STO were mainly carried out by Dr. Keisuke Shibuya *et al.* (Prof. Mikk Lippmaa group) at Tokyo Institute of Technology & University of Tokyo during 2003 to 2007 (the same group; they moved between these two institutes around that time). The primary motivation of them is to achieve high-quality wide-band insulator films on STO with the aim of fabricating all-oxide field-effect transistor (FET).

- 1) In 2003 [“Growth and structure of wide-gap insulator films on SrTiO₃”, Solid-State Electronics 47 (2003) 2211-2214], they reported the growth and structure of epitaxial CaHfO₃/STO, but provided *no* transport data.
- 2) In 2004 [“Domain structure of epitaxial CaHfO₃ gate insulator films on SrTiO₃”, Applied Physics Letters 84 (2004) 2142-2144], they reported the growth of epitaxial CaHfO₃ films on (100) and (110)-oriented STO. Again, they focused on the growth and structure. In this study, they also measured the breakdown field and dielectric constant of the CaHfO₃ films. Note that these measurements are *out-of-plane* across the films, and thus completely different from the *in-plane* transport measurements that characterize the interface conductivity (as in our case). Furthermore, in their measurements of breakdown field and dielectric constant, the CaHfO₃ films were grown on 0.5 wt% Nb-doped STO substrates (the substrates themselves are conducting).
- 3) In 2004 [“Single crystal SrTiO₃ field-effect transistor with an atomically flat amorphous CaHfO₃ gate insulator”, Applied Physics Letters 85 (2004) 425-427; *Ref.*

37 in our original manuscript], they reported that high-quality *amorphous* CaHfO₃ film can be grown on STO, which can work as a high-performance FET. From their data (Fig. 3 & Fig. 4) one can see that, without applying gate bias, the interface is insulating.

- 4) In 2006 [“Field-effect modulation of the transport properties of nondoped SrTiO₃”, Applied Physics Letters 88 (2006) 212116; Ref. 38 in our original manuscript], they reported FET devices with amorphous and epitaxial CaHfO₃ films as gate insulator layers. In the case of epitaxial CaHfO₃ film, the epitaxial CaHfO₃ film has a thickness of ~1.6 nm, and was further covered with a 50-nm amorphous CaHfO₃. The key conclusion is that the insertion of a thin epitaxial CaHfO₃ layer improves the interface quality and thus leads to an enhanced field-effect mobility at low temperature. From their data (Fig. 1 & Fig. 3) one again can see that, without applying gate bias, both kinds of interfaces are insulating. [*In the Point 5 below, the referee raised a possibility to explain why in this work a gate bias was needed to make the interface conducting. We will address this point in details in the response to the Point 5, and further explain why the result in this study is different from our present work.*]
- 5) In 2007 [“Metallic conductivity at the CaHfO₃/SrTiO₃ interface”, Applied Physics Letters 91 (2007) 232106; Ref. 27 in our original manuscript], this time, different from their previous studies, they focused on the interface conductivity and reported that metallic interfaces can be achieved between *amorphous* CaHfO₃ films and STO. Actually, in an earlier work from the same group [“The effect of annealing on SrTiO₃ field-effect transistor devices”, Thin Solid Films 486 (2005) 195-199], they already found that a conducting interface between *amorphous* CaHfO₃ and STO can be achieved without applying gate bias. But in that work their goal is to remove that conductance by annealing in air (The conductance was removed by annealing at 250 °C). In both studies mentioned here, *no* interfaces between epitaxial CaHfO₃ and STO were reported

In short, in all these previous studies, there is *no* claim (the results shown in these studies also *do not* support) that the interfaces between epitaxial CaHfO₃ films and STO are conducting without applying gate bias, while the key finding of our present work is that the interfaces between epitaxial CaHfO₃ films and STO can be conducting without gating. Therefore, we feel that although the *material combination* is not new, *our key finding presented here is new indeed*.

Point 2. “The main differences to previous works of other metal oxide interfaces to STO is the observation that for CaHfO₃/STO, not only does the TiO₂ terminated STO show high conductivity and mobility, but also the SrO terminated STO (other publications have shown an insulating interface for the SrO terminated STO with LaAlO₃ overlayers) and that there is no critical thickness for the CaHfO₃ for the electron gas to be created for both the TiO₂-and SrO-terminated STO.”

Response: We thank the referee for this concise summary. These differences are nontrivial, and make the interfaces of CaHfO₃/STO distinguishing from all other reported metal oxide interfaces.

Point 3. “However, in previous works the critical transition for the metal-insulator transition between 3 and 4 unit cells of overlaying material were observed with in-situ post-annealing in O_2 , which seems not to have been done in this work (Fig. 2a-b). Therefore, the absence of the metal-insulator transition may not be directly comparable to previous findings.

Response: We thank the referee for pointing out this concern. We worry that this concern was unfortunately caused by our failure in clearly presenting our results of post annealing in O_2 . We would like to emphasize that we have carefully performed post annealing in O_2 already, both *in situ* and *ex situ*. While the data shown in Fig. 2a-b are from samples without *in-situ* post annealing in O_2 , we **DID** anneal the same samples *ex situ* (under $P(O_2)=1$ bar, at 500 °C, for 1 hour) and the result was presented in Fig. S3 in the original manuscript (or Fig. S6 in the revised manuscript). As shown in Fig. S6, for $CaHfO_3/STO$ of both terminations, *the 1 and 2 uc samples are still conducting after post annealing in O_2 .*

To further address the referee’s concern, in the revised manuscript we provided additional data for the samples with *in-situ* post annealing in O_2 (here the results of 1, 2, and 4-uc samples were added, in addition to the 8-uc results that have already been presented in Figs. 2c-d in the original manuscript) (see Fig. S7). One can see that *after in-situ post annealing in O_2 the 1 and 2-uc $CaHfO_3/STO$ samples are still conducting.*

Fig. S7 is presented here as Fig. R8.

Fig. R8 | Electrical transport properties of CHO/STO heterointerfaces that have been post annealed in O_2 *in situ*.

Furthermore, we also presented the result for depositing 2-uc (below the critical thickness) $LaAlO_3$ on STO at the same growth conditions as depositing $CaHfO_3$, **without post annealing in O_2** (see Fig. S2). The sample is highly insulating (beyond our measurement limit).

Fig. S2 is re-presented here as Fig. R9.

Fig. R9 | See the blue circle symbols, 2 uc $LaAlO_3/T-STO$.

Therefore, our results demonstrate that, no matter with or without post annealing in O_2 , $CaHfO_3(2\text{ uc})/STO$ is conducting, while $LaAlO_3(2\text{ uc})/STO$ is insulating. This sharp contrast is irrelevant with different post-annealing conditions. **In the revised manuscript, we have improved our presentation to make the results of post annealing in O_2 clearer.** Particularly, we added two sentences in the caption of Fig. 2 to guide readers for the results of post annealing

in O₂.

P10: “The same samples were post annealed *ex situ* in O₂, and the corresponding results are shown in Fig. S6.”

P10: “More *in-situ* annealing results on samples of different thicknesses are presented in Fig. S7.”

Point 4. “The authors present some of the possible reasons for the interface conductivity and claims that the electron gas is of a new nature. However, there is no evidence for that the CaHfO₃/STO interface electron gas has different properties from previous studies, with the exception that also the SrO termination is conductive, which could be caused by oxygen vacancies in the absence of the post-annealing process. (The scenario of oxygen vacancies being responsible for the metallicity is theoretically corroborated ref. 35.)”

Response: We thank the referee for pointing out this concern. We worry that the referee made this comment, again, because of our failure in clearly presenting our results of post annealing in O₂. As stated above in the response to the Point 3, we have actually annealed samples of both terminations in O₂, both *in situ* and *ex situ* (see Figs. 2c-f, S6, & S7). These results clearly demonstrate that we cannot attribute the conductivity of CaHfO₃/SrO-terminated STO to the oxygen vacancies in the absence of the post-annealing process. **In the revised manuscript, we have made the relevant presentations clearer.**

As for the scenario of the electron gas, oxygen vacancies in STO, as proposed in Ref. 35, are one good possibility. However, it might not be the whole story because after post annealing in O₂ in extreme conditions, which should be able to fully remove the oxygen vacancies in STO, the interfaces still have significant remaining conductivity. At this stage, we feel that it is difficult to give a definitive explanation. (Note that in the case of LaAlO₃/STO, the origin of the electron gas is still under hot debate after 15-year intensive studies.) To be conservative, we only presented a few possible reasons for readers. Further studies are needed to fully resolve this question.

Point 5. “The authors further claim that “...an experimental observation of EG at the interface between epitaxial CHO film and STO has never been reported.”, while in ref. 38, field induced conductivity was reported for epitaxial CaHfO₃ grown at somewhat lower temperature (700°C instead of 800°C) and higher oxygen pressure (1 mbar instead of 10⁻⁴ mbar), and lower laser fluency. The direct comparison would imply that the higher oxygen pressure in ref. 38 might have mitigated the oxygen vacancy problem while that interface gets conducting only by gating.”

Response: We thank the referee for pointing out this concern. Again, we worry that the referee made this comment largely because our failure in clearly presenting our results of post annealing in O₂. We stress again that our samples have been post annealed carefully in O₂. In Ref. 38, the sample was post annealed at 400 °C in air for 12 hours; in our work, the samples were post annealed in various conditions, and the most extreme condition is at 1000 °C in 1 bar O₂ flow for 24 hours. In all these cases, the post annealing in O₂/air must have much stronger effect in refilling oxygen vacancies than the effect of detailed growth conditions.

As stated in our response to the Point 1 of the referee, without applying gate bias, no interfacial conductivity, and thus *no EG*, has been reported experimentally yet for the epitaxial CaHfO₃ grown on STO. Although in ref. 38 an interfacial conductivity can be realized by applying an electrical gate bias, this conductivity is induced by field effect and thus obviously different from the EGs that are naturally formed by forming interfaces.

In addition, we would like to figure out one subtle point. In Ref. 38, the real sample configuration is amorphous-CaHfO₃/epitaxial CaHfO₃/STO (see Fig. R10a), rather than CaHfO₃/STO (see Fig. R10b). The thickness of the epitaxial CaHfO₃ is only 4 unit cells while the thickness of the amorphous CaHfO₃ is 50 nm. A couple of recent studies [“Tuning the two-dimensional electron liquid at oxide interfaces by buffer-layer-engineered redox reactions”, Nano Letters 17 (2017) 7062-7066, by *Chen et al.*, Ref. 34 in our manuscript; “Surface amorphous oxides induced electron transfer into complex oxide heterointerfaces”, Advanced Materials Interfaces 5 (2018) 1801216, by our group, Ref. 25 in our manuscript] showed that the presence of the surface amorphous layer can dramatically change the epitaxial film/STO heterointerfaces from insulating to conducting. Therefore, even if the heterointerface in Ref. 38 was conducting (although in fact not), one still cannot conclude that they observed EG at the epitaxial CaHfO₃/STO interface because of the existence of the amorphous layer at the top of the epitaxial CaHfO₃ film in their study.

Fig. R10 | Schematic drawing of the sample configuration in (a) Ref. 38 and (b) our present work.

Therefore, our present work should be the first experimental observation of EG at the interface between epitaxial CaHfO₃ films and STO substrates. However, to avoid any potential misleading, in the revised manuscript (page 3) we are happy to modify the initial statement to “Here we show that the interface between epitaxial CHO and STO can be metallic as well.”

Point 6. “In conclusion, I see the main benefits of the presented work to be an additional observation of what has been reported before, but now also with SrO-terminated STO (although for different growth conditions) and a claimed robustness of the interface. For this reason, I would recommend that the paper would find a more appropriate journal in e.g., Journal of Applied Physics or Applied Physics Letters.”

Response: We respectfully disagree with the referee about this comment. As quoted from the other three referees, our present work is of high novelty, and could be important in both fundamental physics of oxide heterointerfaces and device applications.

Referee #1: “...This is important news to the field, and of possible relevance for applications...”.

Referee #2: “...These results are potentially very interesting and intriguing...”.

Referee #3: “...I view this manuscript is timing and very interesting...More importantly, identifying a system which is robust to oxygen annealing and surface dopants has significant importance for practical application...”.

To address the referee’s concerns, we have provided new experimental data and improved our presentation. Now it should be clear that oxygen vacancies induced by oxygen-deficient growth condition should not be a worried issue *because we have carefully post annealed our samples in oxygen-rich environments* (Note that one of our key findings is the robustness to annealing in oxygen.). It should also be clear that the observation of electron gas at the epitaxial CaHfO₃/SrTiO₃ heterointerfaces is new, although the material combination is similar with previous studies.

We hope now the referee can be satisfied with our revised manuscript, and agrees with its publication in Nature Communications.

Point 7. “Additional comments to the text:

- 1) p.2 l.1: “SrTiO₃ (STO) is the workhorse oxide semiconductor.” Maybe write “SrTiO₃ (STO) is the oxide semiconductor workhorse.” Or “SrTiO₃ (STO) is the workhorse of oxide semiconductors.”
- 2) p.2 l.6: Maybe write “...superconductivity at the interface...”
- 3) p.2 l.19: Maybe write “that the EGs can be generated...”
- 4) p.4 Fig. 1: l.6 Maybe write “g, h, Cross-sectional STEM images of 8 uc...”
- 5) p.9 Fig. 3: Maybe write “The top gate electrode is a 100-nm-thick silver film.” (Misspelled “top” and missing “a”.)
- 6) p.9 l.14-15: Maybe write “...to achieve sharp step-and-terrace surfaces” or “...to achieve a sharp step-and-terrace surface”
- 7) Check also page reference of ref. 35 (Appl. Phys. Lett. 98, 133114 (2011))!”

Response: We sincerely thank the referee for kindly pointing out these problems in our manuscript. **We have corrected all of them in the revised manuscript.**

General information: In the revised manuscript, we wrote all the changes in RED.

REVIEWERS' COMMENTS:

Reviewer #2 (Remarks to the Author):

I have reviewed the revised manuscript that describes a novel electron gas at the CFO/STO interface. I find the revisions to adequately address my concerns (and those of the other referees). I can now recommend publication in Nature Communication.

Reviewer #3 (Remarks to the Author):

The authors have clearly answered my questions. I also agree with the authors that the results of the epitaxial CaHfO₃/STO interface are new, original and interesting, and I recommend the current manuscript for publication in Nature Communications without any delay.

Reviewer #4 (Remarks to the Author):

In this second version and with the answers to the referees, it seems to me that the manuscript has become more transparent and substantiated. The direct comparative studies using identical conditions with LAO/STO and bare STO further corroborate the notion that the CHO/STO interface possess different properties compared to other interfaces, and that conditions similar to earlier works on LAO/STO was clarified in the rebuttal letter and the manuscript to support their claims. Even though it remains to be seen if the findings will change the playground for oxide electronics, I feel that the paper may now meet the conditions for publishing in Nature Communications.